# The Preventive Effect of the Phenotype of Tumour-Associated Macrophages, Regulated by CD39, on Colon Cancer in Mice

**DOI:** 10.3390/ijms22147478

**Published:** 2021-07-13

**Authors:** Hyun-Jun Park, Eun-Hye Seo, Liyun Piao, Sang-Tae Park, Min-Ki Lee, Seong-Eun Koh, Seung-Hyun Lee, Seong-Hyop Kim

**Affiliations:** 1Korea Radioisotope Center for Phamaceuticals, Korea Institute of Radiological and Medical Sciences, Seoul 01812, Korea; parkhyunj111@naver.com; 2BK21 Plus, Department of Cellular and Molecular Medicine, Konkuk University School of Medicine, Seoul 05029, Korea; gmreo@naver.com; 3Department of Infection and Immunology, Konkuk University School of Medicine, Seoul 05029, Korea; piaoliyun625@naver.com; 4Department of Anesthesiology and Pain Medicine, Konkuk University Medical Center, Konkuk University School of Medicine, Seoul 05030, Korea; 20190151@kuh.ac.kr (S.-T.P.); 20200352@kuh.ac.kr (M.-K.L.); 5Department of Rehabilitation Medicine, Konkuk University Medical Center, Konkuk University School of Medicine, Seoul 05030, Korea; gohse@kuh.ac.kr; 6Department of Microbiology, Konkuk University School of Medicine, Seoul 05029, Korea; shlee@kku.ac.kr; 7Department of Medicine, Institute of Biomedical Science and Technology, Konkuk University School of Medicine, Seoul 05029, Korea

**Keywords:** CD39, CD73, tumour-associated macrophage, colon cancer

## Abstract

Background: This study was designed to investigate the effect of cluster differentiation (CD)39 and CD73 inhibitors on the expresion of tumour-associated macrophages (TAMs), M1- versus M2-tumour phenotypes in mice with colon cancer. Methods: An in vitro study of co-culture with colon cancer cells and immune cells from the bone marrow (BM) of mice was performed. After the confirmation of the effect of polyoxotungstate (POM-1) as an inhibitor of CD39 on TAMs, the mice were randomly divided into a control group without POM-1 and a study group with POM-1, respectively, after subcutaneous injection of CT26 cells. On day 14 after the injection, the mice were sacrificed, and TAMs were evaluated using fluorescence-activated cell sorting. Results: In the in vitro study, the co-culture with POM-1 significantly increased the apoptosis of CT26 cells. The cell population from the co-culture with POM-1 showed significant increases in the expression of CD11b^+^ for myeloid cells, lymphocyte antigen 6 complex, locus C (Ly6C^+^) for monocytes, M1-tumour phenotypes from TAMs, and F4/80^+^ for macrophages. In the in vivo study, tumour growth in the study group with POM-1 was significantly limited, compared with the control group without POM-1. The expressions of Ly6C^+^ and major histocompatibility complex class II^+^ for M1-tumour phenotypes from TAMs on F4/80^+^ from the tumour tissue in the study group had significantly higher values compared with the control group. Conclusion: The inhibition of CD39 with POM-1 prevented the growth of colon cancer in mice, and it was associated with the increased expression of M1-tumour phenotypes from TAMs in the cancer tissue.

## 1. Introduction

Tumour-associated macrophages (TAMs) are unique tumour-infiltrating leukocytes that directly affect tumour progression, according to the anti (M1) or pro (M2) tumour phenotype. Macrophages indirectly affect tumour progression by acting as antigen-presenting cells that trigger secondary immune responses by stimulating naive T cells to become effector cells [1,2,3]. TAMs are abundantly distributed in the environment of solid tumours and play a pivotal role in tumour progression [4]. Moreover, TAMs are associated with chemotherapy resistance in solid tumours and have been used as one of the therapeutic targets [5]. Therefore, the proper distribution of the M1-polarized phenotype of macrophages is required to prevent tumour progression. However, the roles of TAMs in colon cancer have produced controversy [6,7].

Cluster of differentiation (CD)39 and CD73, known as ectonucleotidases, hydrolyse adenosine triphosphate (ATP) into adenosine and have been reported to regulate cancer progression [8,9,10,11]. Extracellular ATP stimulates effector immune cells and activates the anti-cancer immune response. In contrast, extracellular adenosine suppresses effector immune cells and promotes cancer progression. Therefore, regulation of the ATP-to-adenosine cascade by CD39 and CD73 plays an important role in cancer suppression and progression [12,13,14,15,16]. Regulating the ATP-to-adenosine cascade by CD39 and CD73 may affect the distribution of TAMs. However, the effects of CD39 and CD73 on the TAM phenotype have not been investigated.

We hypothesized that regulation of the ATP-to-adenosine cascade by CD39 and CD73 is associated with the TAM phenotype. Inhibiting CD39 and CD73 may increase expression of the M1 TAM phenotype and prevent cancer progression. This study was designed to investigate the effects of CD39 and CD73 inhibitors on the expression of the M1 and M2 TAM phenotypes in mice with colon cancer.

## 2. Results

### 2.1. In Vitro Study

Co-culture of CT26 and bone marrow (BM) cells resulted in greater Annexin V expression in CT26 cells, compared with CT26 cells cultured alone (5.26 ± 0.90% vs. 2.12 ± 0.98%, *p* < 0.01) (Figure 1B), but lower Annexin V expression in BM cells, compared with BM cells cultured alone (30.74 ± 3.72% vs. 46.41 ± 3.74%, *p* < 0.01) (Figure 1C). Treatment of the co-cultures with polyoxotungstate (POM-1) resulted in significant differences in Annexin V expression in CT26 and BM cells, compared with no POM-1 treatment (CT26 cells, 7.15 ± 0.88% vs. 5.26 ± 0.90%, *p* < 0.01; BM cells, 13.34 ± 2.37% vs. 30.74 ± 3.72%, *p* < 0.01) (Figure 1D). However, treatment with adenosine 5’-(α,β-methylene)diphosphate (APCP) did not affect Annexin V expression (Figure 1E).

The effect of POM-1 on the cell populations in BM and CT26 co-cultures was evaluated. The expression levels of CD11b in myeloid cells (Figure 2B), lymphocyte antigen 6 complex, locus C (Ly6C) in monocytes and M1 TAMs (Figure 2C), and F4/80 in macrophages (Figure 2D) were increased significantly in the BM and CT26 co-cultured cells in the presence of POM-1 compared with the absence of POM-1 (CD11b, 45.15 ± 1.95% vs. 41.96 ± 1.38%, *p* < 0.05; Ly6C, 1.20 ± 0.21% vs. 0.78 ± 0.15%, *p* < 0.05; F4/80, 0.59 ± 0.12% vs. 0.20 ± 0.07%, *p* < 0.001). However, major histocompatibility complex (MHC) class II expression did not differ in the BM and CT26 co-cultures according to POM-1 treatment (Figure 2E).

Based on these results, the CD39 inhibitor POM-1 was chosen to evaluate the effect of regulating CD39 and CD73 on the M1 versus M2 tumour phenotype in vivo.

### 2.2. In Vivo Study

Ten mice were evenly allocated into the control (saline treatment) and study (POM-1 treatment) groups. No complications occurred in either group. The tumour volume at 14 days after treatment with saline/POM-1 was significantly greater in the control group than the study group (280.43 ± 133.57 vs. 124.41 ± 50.33 mm^3^, *p* < 0.05) (Figure 3).

The expression of Ly6C and MHC class II in F4/80^+^ tumour tissues was significantly higher in the study group than the control group (Ly6C, 28.16 ± 3.23% vs. 15.30 ± 8.30%, *p* < 0.05; MHC class II, 69.48 ± 17.67% vs. 46.64 ± 7.56%, *p* < 0.05) (Figure 4A). The expression of MHC class II in F4/80^+^ tumour-draining lymph nodes (TDLNs) was significantly lower in the study group than the control group (26.02 ± 9.93% vs. 51.44 ± 11.47%, *p* < 0.01) (Figure 4B). The expression of TAM markers was not significantly different between the groups. CD8 expression in the spleen was significantly higher in the study group than the control group (16.51 ± 2.60% vs. 10.43 ± 4.47%, *p* < 0.05) (Figure 4C). The expression of CD4 in blood was significantly higher in the study group than the control group (35.93 ± 5.22% vs. 23.61 ± 7.65%, *p* < 0.05) (Figure 4D), but TAM marker expression was unaffected. Caspase-3 expression was significantly higher in the study group than the control group (9.56 ± 2.45% vs. 7.40 ± 1.96%, *p* < 0.01), whereas the expression of F4/80 was similar between the groups (2.64 ± 1.86% vs. 2.66 ± 1.09%, *p* = 0.72) (Figure 5).

## 3. Discussion

In the in vitro study, the CD39 inhibitor POM-1 increased the numbers of myeloid cells, monocytes, and TAMs with the M1 phenotype. The in vivo study of mice with colon cancer demonstrated that POM-1 prevented tumour growth. POM-1 was associated with increased expression of the M1 TAM phenotype in cancer tissues.

Regulation of the cascade from ATP to adenosine by CD39 and CD73 during cancer growth has been associated with direct or indirect activation of specific immune cells. Considering that the intercellular calcium signal induced by ATP potentiates the phagocytosis of macrophages [17], regulation of the ATP-to-adenosine cascade by CD39 and CD73 is expected to play an important role in macrophages in cancer growth. Cohen et al. reported that CD39 activates the hydrolysis of ATP and inhibits the activation of macrophages [18]. Moreover, the ATP-to-adenosine cascade by CD39 and CD73 controls the activation of regulatory T cells in cancer [19]. The activation of regulatory T cells has immunomodulatory functions in other immune cells, including macrophages [20].

POM-1 was used as the CD39 inhibitor in the present study. POM-1 is an inorganic metal oxide that inhibits enzymes with important roles in cancer growth. Therefore, it is expected to be a next-generation anti-cancer drug [21,22,23]. This is the first report of an increased expression of the M1 TAM phenotype by POM-1 treatment.

The CD73 inhibitor APCP did not affect the differentiation of TAMs due to the low expression of CD73, compared with CD39, in TAMs. Lévesque et al. did not detect CD73 expression on macrophages in mice [24], and Zanin et al. did not detect hydrolysis of AMP to adenosine in macrophages in mice [25], indicating that the effect of CD73 on TAMs is limited. Therefore, APCP as a CD73 inhibitor does not have any effect on TAMs.

Interestingly, the expression of MHC class II in F4/80^+^ TDLNs decreased significantly after POM-1 treatment in the in vivo study, which may be associated with the recruitment of TAMs to the tumour. We also confirmed the presence of CD4^+^ and CD8^+^ T cells in vivo to evaluate the secondary immune response. Although POM-1 treatment caused no significant change in TAMs, it induced significantly higher values of CD8^+^ T cells in the spleen and CD4^+^ T cells in the blood. The proliferation of CD4^+^ and CD8^+^ T cells was observed in the spleen and blood in colon cancer under increased ATP levels after inhibition of CD39 [26]. Kashyap et al. reported that changes in T cells resulting from increased ATP levels, with changes in CD39 expression in TAMs, were observed in mice with colon cancer after treatment with antisense oligonucleotides suppressing CD39 expression [27]. Therefore, reciprocal effects between TAMs and CD4^+^ or CD8^+^ T cells are expected in the cancer environment, but further investigations are needed for clarification.

In the present study, we focused on the differentiation of M1 TAMs after POM-1 treatment in cancer. Therefore, we evaluated Ly6C expression in F4/80^+^ cancer tissues as the primary outcome and MHC class II expression as the secondary outcome because both Ly6C and MHC class II are specific markers of the M1 TAM phenotype.

Myeloid progenitors derived from multi-potential hematopoietic stem cells differentiate into myeloblasts, which in turn differentiate into granulocytes and monocytes, which in turn differentiate into macrophages [28,29]. The significant increase in CD11b expression after POM-1 treatment in the in vivo study was expected. Although CD11b expression was not verified in the in vivo study to focus on the M1 TAM phenotype, the evaluation of other immune cells differentiated from myeloid cells would be helpful to determine the direct or indirect reciprocal effects of TAMs and other immune cells in colon cancer.

Pimenta-Dos-Reis reported that POM-1 inhibits activation of macrophages in mice with sepsis induced by lipopolysaccharide and reduces the levels of related cytokines [30]. This finding indicates that the effect of POM-1 on the activation or differentiation of TAMs may differ according to the condition. In the present study, POM-1 was administered systemically, rather than focally, via intraperitoneal injection. Therefore, the effect of POM-1 on tumour tissues would be insufficient, and immune cell cascades would be different, compared with those induced by focal administration.

In conclusion, inhibiting CD39 with POM-1 prevented the growth of colon tumours in mice and was associated with induction of the TAM M1 phenotype in cancer tissues.

## 4. Materials and Methods

We confirmed the effects of the CD39 and CD73 inhibitors on the M1 TAM phenotype in vivo in TAMs co-cultured with a colon cancer cell line. An in vivo study was performed in mice with colon cancer.

### 4.1. Animals

These experiments were approved by, and performed following the guidelines of, the Institutional Animal Care and Use Committee of Konkuk University (approval number: KU19034; approval date: 20 March 2019) and were conducted at the Konkuk University Laboratory Animal Research Centre. Six- to eight-week-old Balb/c mice were purchased from ORIENT BIO Inc. (Seongnam, Korea) and were housed at the Konkuk University Laboratory Animal Research Centre.

### 4.2. In Vitro Co-Cultures of Colon Cancer and Immune Cells, Including Macrophages

An in vitro study was performed to confirm the effects of CD39 and CD73 on immune cells, including macrophages, in colon cancer. 

### 4.3. Preparation of Colon Cancer Cell Line

The CT26 murine colon cancer cell line derived from Balb/c mice was purchased from the American Type Culture Collection (Manassas, VA, USA). The cells were cultured in complete medium containing 10% foetal bovine serum (FBS) and 1% penicillin/streptomycin in RPMI-1640 medium (Thermo Fisher Scientific, Waltham, MA, USA). The CT26 cells were harvested with trypsin-EDTA (Thermo Fisher Scientific) and then counted. Cell viability was confirmed to be >95% by the trypan blue exclusion assay (Sigma-Aldrich, St. Louis, MO, USA).

### 4.4. Preparation of Immune Cells

Immune cells, including myeloid cells, monocytes, macrophages, and M1 TAMs, were collected from BM. BM was obtained from the femur and tibia of mice and harvested after perfusion of the bones with culture medium. The cells from the perfused BM were filtered through a 70-μm-pore cell strainer (Corning, Corning, NY, USA) and resuspended in red blood cell lysis buffer (Sigma-Aldrich). After washing in phosphate-buffered saline (PBS), the cells were counted. Cell viability was confirmed to be >95%.

### 4.5. Co-Culture

The CT26 and BM cells were co-cultured at a 1:20 ratio. The CT26 and/or BM cells were co-cultured with or without a CD39 inhibitor (POM-1; 30 μM, Sigma-Aldrich) or CD73 inhibitor (APCP; 30 μM, Sigma-Aldrich) for 24 h.

### 4.6. Cells Analyses Including Apoptosis

CT26 and immune cells were identified by flow cytometry using forward and side scatter (FSC and SSC) gating. The CT26 cells were defined by the high FSC and SSC areas and the immune cells by the low FSC and SSC areas within a sample plot. The immune cell markers used were CD11b for myeloid cells, Ly6C for monocytes and M1 TAMs, F4/80 for macrophages, and MHC class II for M1 TAMs. Apoptosis of CT26 and immune cells after co-culture was determined according to Annexin V expression. The co-cultured cells were washed with staining buffer containing 0.1% FBS and 0.05% sodium azide in PBS and then stained with antibodies against the various markers in staining buffer. After washing with staining buffer, the cells were analysed by fluorescence-activated cell sorting (FACS) (Becton-Dickinson, Franklin Lakes, NJ, USA).

### 4.7. In Vivo Study

#### 4.7.1. Establishment of the Colon Cancer Model and Harvesting of Tissues

Based on the in vivo co-culture study, the CD39 inhibitor POM-1 was used to inhibit CD39 in vivo. A 50 μL aliquot of CT26 cells (5.0 × 10^5^ cells) in PBS was subcutaneously injected into the shaved right flank. The mice were randomly divided into the control and study groups. The control group was intraperitoneally injected with 50 μL normal saline and the study group with 50 μL POM-1 (5 mg/kg) daily for 2 weeks. The growth and tumour volume of the subcutaneously injected CT26 cells was recorded daily for 2 weeks by determining the length and width using callipers. Tumour volume was calculated as (length × width^2^)/2. After confirming tumour growth at 14 days, the mice were sacrificed using inhaled isoflurane and then underwent cervical dislocation. The abdomens, including the flanks of the mice, were sterilized with 70% ethanol and opened. Whole blood was collected via heart puncture and stored in EDTA-coated tubes. The whole blood was diluted with PBS and loaded on 1 mL Ficoll-Hypaque (Sigma-Aldrich) to isolate the mononuclear cells by centrifugation. The isolated mononuclear cells were washed with PBS and stored in staining buffer before FACS. The tumours, TDLNs, and spleens were harvested, minced into small pieces, and dissociated using collagenase type IV (Sigma-Aldrich). The dissociated tissues were filtered through a 70 μm-pore cell strainer, and the filtered cells were washed and stored in staining buffer before FACS.

#### 4.7.2. Populations of TAMs and Other Immune Cells in the Tumour and Organs

Flow cytometry was used to identify immune cells in the tumours, TDLNs, spleen, and blood based on the following markers: Ly6C for monocytes and M1 TAMs, F4/80 for macrophages, MHC class II for M1 TAMs, CD4 for CD4^+^ T cells, and CD8 for CD8^+^ T cells. The immune cells were washed with staining buffer, stained with antibodies against the markers in staining buffer, washed again with staining buffer, and analysed by FACS.

### 4.8. Colon Cancer Immunohistochemistry

The rates of apoptosis in the cancer cells and macrophages were determined. The harvested tumour tissues were fixed in 4% paraformaldehyde solution for 1 day (Biosesang, Seongnam, Korea), washed in tap water, and processed (Leica Biosystems, Wetzlar, Germany). The fixed specimens were embedded in a paraffin block on the embedding centre (Leica Biosystems) and sectioned to a thickness of 4–5 μm using a microtome (Leica Biosystems). The sections were loaded onto silane-coated glass slides (Muto, Tokyo, Japan). The slides were baked overnight at 40 °C and rehydrated by sequential immersion in xylene and a graded ethanol series (100%, 90%, 80%, and 70%). The slides were boiled with citrate buffer to retrieve the antigen in the fixed samples. The slides were treated with 3% hydrogen peroxide solution to inhibit endogenous peroxidase activity, placed in a humid chamber, and stained with anti-caspase-3 antibody to visualize apoptosis of the cancer cells and F4/80^+^ macrophages. The slides were then washed and treated sequentially with an avidin–biotin complex reagent (Vector Laboratories, Burlingame, CA, USA) and 3,3’-diaminobenzidine substrate (Vector Laboratories) to visualize staining. The slides were counterstained with haematoxylin (Sigma-Aldrich) and dehydrated through a graded ethanol series (70%, 80%, 90%, and 100%), followed by xylene. The slides were covered with mounting solution (Vector Laboratories) and cover slipped. Images were taken under a microscope (Leica Biosystems). The expression of caspase-3 and F4/80 in tumour tissues was analysed using the ImageJ program (Wayne Rasband, National Institutes of Health, Bethesda, MD, USA).

### 4.9. Statistical Analysis

The primary and secondary outcomes of the study were the expression of Ly6C and MHC class II, respectively, in F4/80^+^ cancer tissues. A pilot study conducted in six mice showed that the Ly6C expression level in F4/80^+^ tissues was 14.5 ± 5.4% in the control group (n = 3) and 25.2 ± 3.0% in the study group (n = 3), respectively. In the same pilot study, the expression level of MHC class II in F4/80^+^ tissues was 43.65 ± 7.40% in the control group and 70.12 ± 12.80% in the study group. Based on the pilot study, 10 and 8 mice were needed to evaluate the primary and secondary outcomes, respectively, with an α of 0.05 and power of 0.9.

The difference in tumour growth between the groups was analysed by two-way repeated-measures analysis of variance. Differences in the other variables between the groups were analysed by *t*-test or Mann–Whitney U test after confirmation of normality, using Shapiro–Wilk test. GraphPad Prism software (ver. 5.01; La Jolla, CA, USA) was used for the statistical analysis. Data are presented as mean ± standard deviation. *p*-values < 0.05 were considered significant.

## Figures and Tables

**Figure 1 ijms-22-07478-f001:**
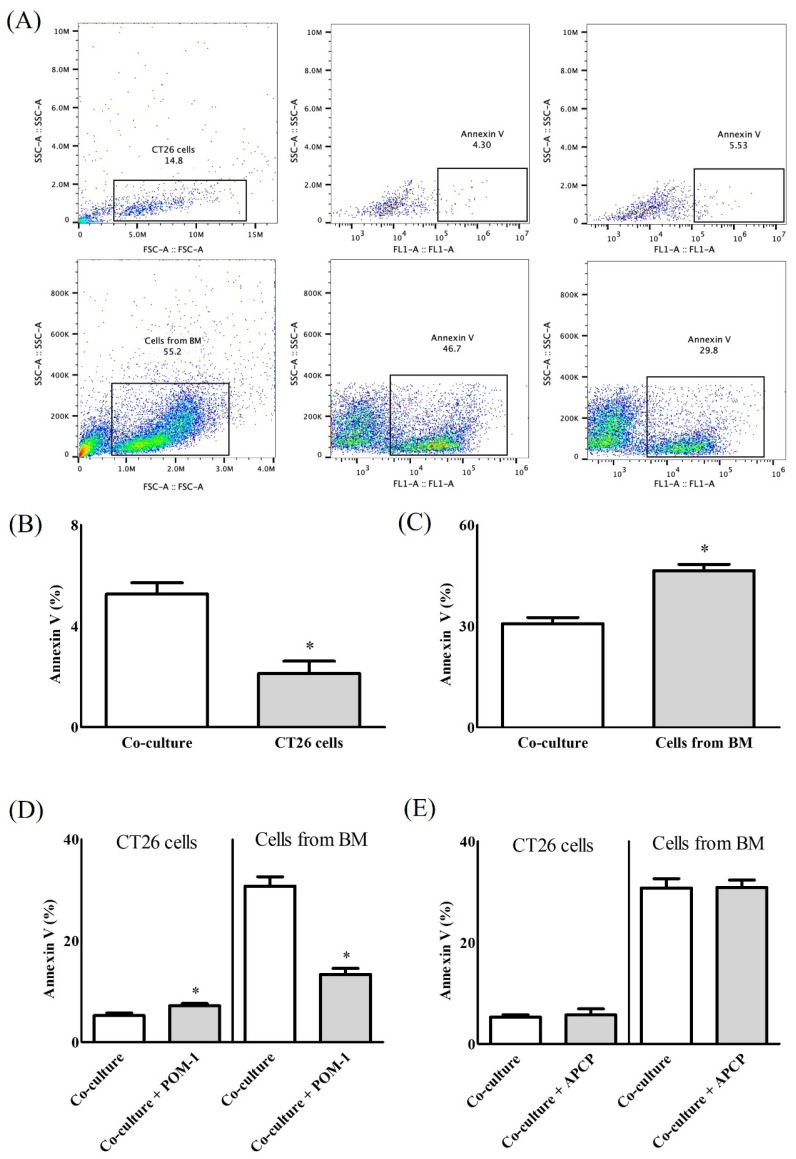
Apoptosis of co-culture of CT26 cells and the cells from bone marrow (BM). (**A**) Representative image of gating strategy for flow cytometry, (**B**) Apoptosis of CT26 cells, (**C**) Apoptosis of the cells from BM, (**D**) Apoptosis at co-culture with polyoxotungstate (POM-1) as an inhibitor of cluster differentiation (CD)39, and (**E**) Apoptosis at co-culture with adenosine 5’-(α,β-methylene)diphosphate (APCP) as an inhibitor of CD73. *: *p* < 0.05.

**Figure 2 ijms-22-07478-f002:**
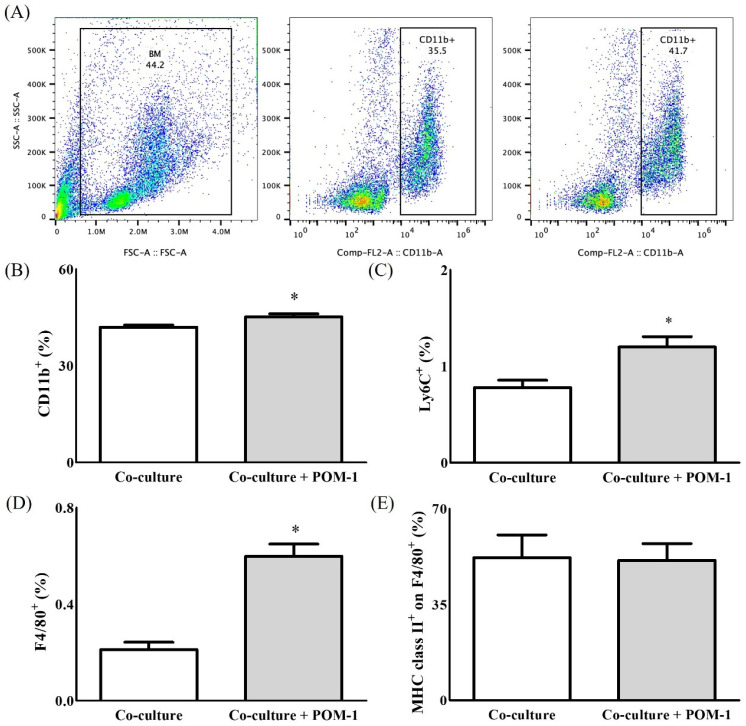
The expressions of immune cells in co-culture with/without polyoxotungstate (POM-1) as an inhibitor of cluster differentiation (CD)39. (**A**) Representative image of gating strategy for flow cytometry, (**B**) The expression of CD11b^+^ for myeloid cells, (**C**) The expression of lymphocyte antigen 6 complex, locus C (Ly6C^+^) for monocytes and M1-tumour phenotypes from as tumour-associated macrophages (TAMs), (**D**) The expression of F4/80^+^ for macrophages and (**E**) The expression of major histocompatibility complex (MHC) class II^+^ for M1-tumour phenotypes from TAMs. *: *p* < 0.05.

**Figure 3 ijms-22-07478-f003:**
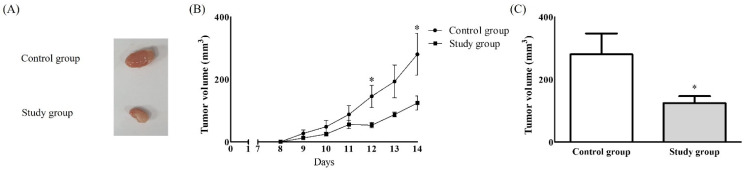
Tumour growth after subcutaneous injection of CT26 cells. (**A**) Tumour, (**B**) Tumour volume according to the time, (**C**) Tumour volume on 14 days after the injection. *: *p* < 0.05.

**Figure 4 ijms-22-07478-f004:**
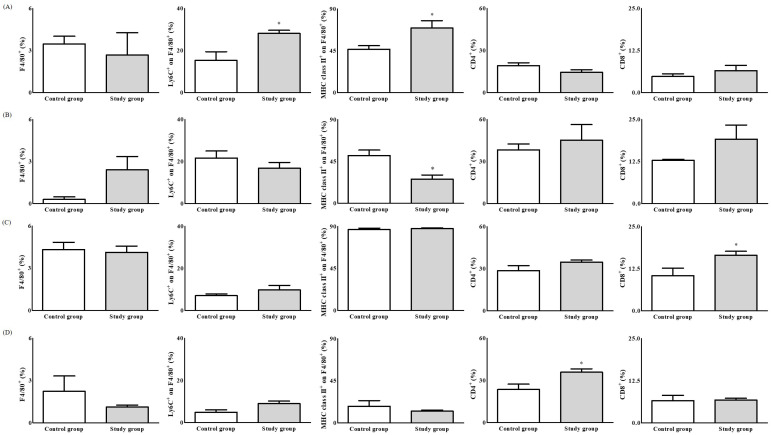
The expressions of F4/80^+^ for macrophages, lymphocyte antigen 6 complex, locus C (Ly6C^+^) for monocytes, and M1-tumour phenotypes from tumour-associated macrophages (TAMs) on F4/80^+^, major histocompatibility complex (MHC) class II^+^ for M1-tumour phenotypes from TAMs on F4/80^+^, cluster differentiation (CD)4^+^ for CD4^+^ T cells, and CD8^+^ for CD8^+^ T cells. (**A**) Tumour tissue, (**B**) Tumour-draining lymph node (TDLN), (**C**) Spleen, and (**D**) Blood. *: *p* < 0.05.

**Figure 5 ijms-22-07478-f005:**
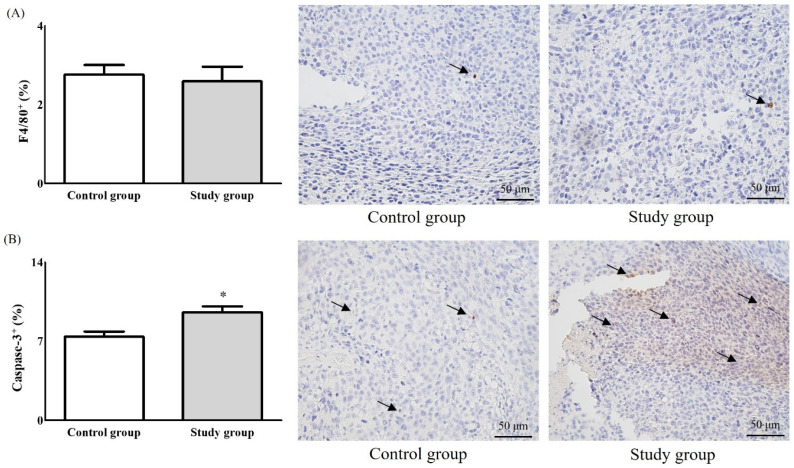
Immunohistochemistric analysis with the expressions of F4/80^+^ (**A**) for macrophages and caspase-3 (**B**) for cancer cells. *: *p* < 0.05. The arrows means “F4/80+” in Figure 5A and “Caspase-3” in Figure 5B, respectively.

## Data Availability

The data presented in this study are available on request from the corresponding author.

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
