# Peer review of "The Preventive Effect of the Phenotype of Tumour-Associated Macrophages, Regulated by CD39, on Colon Cancer in Mice"

_ijms, 2021, doi:10.3390/ijms22147478_

Round 1
Reviewer 1 Report
In this paper entitled “The effect of the phenotype of tumour-associated macrophages, regulated by cluster differentiation (CD)39, on colon cancer in mice” Hyun-Jun Park et al investigated the effect for inhibitor of cluster differentiation (CD)39 and CD73 on the expression of tumour-associated macrophages between M1- and M2-tumour phenotypes in mice suffering from colon cancer.
The work is very interesting, the technical approach is well described, and data obtained interesting. The authors found that the inhibition of CD39 with POM-1 prevented the growth of colon
cancer in mice and it was associated with the increased expression of M1-tumour phenotypes from TAMs in the cancer tissue. The results are well argumented in the final discussion.
Reviewer 2 Report
The article entitled: "The effect of the phenotype of tumour-associated macrophages, regulated by cluster differentiation (CD)39, on colon cancer in mice" by Park et al., provides a rational for the use of inhibitors of CD39 against colorectal cancer progression. The manuscript is overall well written, although it needs some revision for English grammar, style and usage. M1- and M2-macrophages are extremely crucial for cancer initiation and progression; however, introduction is poor and must be extended, e.g.: concerning the role of M1 and M2 in cancer in general and in colorectal cancer in particular. Some other points need to be addressed to improve the quality of the manuscript.
Minors
I suggest to remove "cluster differentiation" from the Title and leave it as: "The effect of the phenotype of tumour-associated macrophages, regulated by CD39 on colon cancer in mice". The Title could attrack more attention whether author define better which "effect" they observed.
In sentence: "(CD)39 and CD73 on the expression of tumour-associated macrophages (TAMs)" from the abstract section, please rephrase this statement since TAMs are not expressed, they are present or enriched.
In vivo and in vitro are latin terms; thus, they must be written in cursive.
Cells lines must be placed without (parentheses).
In M&M section "2.2. In vitro co-cultures of colon cancer and immune cells, including macrophages" several methods have been included here: cell lines and culture, co-culture, cell viability, cytometry or apoptosis. Please differentiate each of the methods individually.
All "et al.," expressions must be written in cursive since they are Latin expressions, and followed by point since they are abbreviation from "et alii".
Majors
Please justify whether you are working with parametric or non-parametric varibles, and then explain why Mann-Whitney test has not been used for this purpose.
Figure 1 must me accompanied by FSC-SSC Scatterplot for FACS.
Figure 2 must also show some representative cytometry images.
Figure 3. Please show a picture from mice tumor treatet and control.
Figure 3. How is it possible that Figure 3b and 3c are just the same while one is tumor volume and the other is tumor weight? There is any linear correlation between tumor weight and volume?
Figure 5. Please include antigens or antibody for the IHC.
Round 2
Reviewer 2 Report
Thanks for take into consideration all my suggestions. In my opinion the article deserves publication